# The Effect of Exit Time and Entropy on Asset Performance Evaluation

**DOI:** 10.3390/e25091252

**Published:** 2023-08-23

**Authors:** Mohammad Ghasemi Doudkanlou, Prokash Chandro, Shokoofeh Banihashemi

**Affiliations:** 1Department of Economics and Statistics, University of Siena, 53100 Siena, Italy; 2Department of Accounting and Finance, Turku School of Economics, The University of Turku, 20500 Turku, Finland; prokash.p.chandro@utu.fi; 3Department of Mathematics, Allameh Tabataba’i University, Tehran 1489684511, Iran; shbanihashemi@atu.ac.ir

**Keywords:** portfolio performance evaluation, data envelopment analysis, Shannon entropy, stop strategy, risk management, CVaR, SPP-CVaR

## Abstract

The objective of this study is to evaluate assets’ performance by considering the exit time within the risk measurement framework alongside Shannon entropy and, alternatively, excluding these factors, which can be used to create a portfolio aligned with short- or long-term objectives. This portfolio effectively balances the potential risks and returns, guiding investors to make decisions that are in line with their financial goals. To assess the performance, we used data envelopment analysis (DEA), whereby we utilized the risk measure as an input and the mean return as an output. The stop point probability–CVaR (SPP-CVaR) was the risk measurement used when considering the exit time. We calculated the SPP-CVaR by converting the risk-neutral density to the real-world density, calibrating the parameters, running simulations for price paths, setting the stop-profit points, determining the exit times, and calculating the SPP-CVaR for each stop-profit point. To account for negative data and to incorporate the exit time, we have proposed a model that integrates the mean return and SPP-CVaR, utilizing DEA. The resulting inefficiency scores of this model were compared with those of the mean-CVaR model, which calculates the risk across the entire time horizon and does not take the exit time and Shannon entropy into account. To accomplish this, an analysis was conducted on a portfolio that included a variety of stocks, cryptocurrencies, commodities, and precious metals. The empirical application demonstrated the enhancement of asset selection for both short-term and long-term investments through the combined use of Shannon entropy and the exit time.

## 1. Introduction

The stop strategy integrates components of both the stop-loss and take-profit strategies. The stop-loss strategy limits losses by selling a security when it reaches a predetermined price level below the current market price, while the take-profit strategy locks in profits by selling a security at a predetermined price level above the current market price (Eiteman, 1966) [1]. Numerous scholarly works have been dedicated to examining the concept of the stop strategy, such as Tschoegl (1988), Osler (2003), and Bensaid and Olivier (2000) [2,3,4]. However, the exit time remains uncertain because investors cannot predict when the price will reach the stopping point due to unpredictable market fluctuations.

The stop strategy is a popular risk management technique used by traders and investors in the stock market. Research conducted by Osler (2003) indicated that the clustering of stop-loss and take-profit orders can help elucidate certain patterns observed in financial markets [3]. These patterns are commonly utilized by traders in technical analysis. Investors using hedge portfolios and the stop strategy to trade may find that traditional risk measures are not adequate for accurately measuring the risk of their portfolios. This is because hedge portfolios and stop strategies are designed to offset risks and limit losses, which can result in missed opportunities and increased transaction costs. Investors using stop strategies are often concerned with limiting their losses, and the stop point is set as a way to achieve that.

However, since the exit time is uncertain and can be influenced by market conditions, the investor’s focus is on managing risk before reaching the stop point. Once the transaction is closed, the risk measures become irrelevant because the investor is no longer exposed to the market. Traditional risk measures evaluate risk over the entire time horizon, which does not account for the uncertainty in the exit time caused by the stop strategy. Therefore, alternative risk measures may be needed to better capture the dynamics of managing risk with a stop strategy.

Entropy, a term first introduced in 1865 by the German physicist Rudolf Clausius for studying thermodynamics, has been used to gauge order, disorder, and uncertainty across diverse disciplines. Its initial application in finance, pioneered by Philippatos and Wilson for portfolio selection [5], has since broadened to encompass asset pricing and option pricing, solidifying its role as a key tool in financial research and decision-making. Shannon entropy can be applied to measure the degree of uncertain of a portfolio’s return [5]. However, it can also be used to measure the degree of diversification of investments [6].

DEA is a nonparametric linear programming technique used to assess the efficiency and productivity of decision-making units (DMUs). Originally introduced as a managerial and performance measurement tool in the late 1970s, DEA has since seen extensive application and adaptations across diverse fields and industries. Notably, it has gained significance in portfolio and asset pricing research, where it aids in measuring the relative efficiencies and improving portfolios’ performance (Eberlein and Keller, 2023) [7].

The conventional DEA models typically assume that all inputs and outputs are non-negative; however, in the case of asset returns, both positive and negative values may be encountered over a certain period. As a solution to this challenge, we utilized the range directional measure (RDM) model, which has been specifically designed to address negative rates of return, as recommended by Portela et al. (2004) [8].

Markowitz (1952) developed the modern portfolio theory, which utilizes the expected return and risk to optimize portfolios and diversify investments [9]. He introduced the mean-variance model to consider an asset’s anticipated return and risk. Various studies have investigated the optimization of stock portfolios using the mean-variance model introduced by Merton (1969, 1971), Magill and Constantinides (1976), and Davis and Norman (1990) [10,11,12,13]. However, the model has faced criticism due to its reliance on variance as a measure of risk.

The SPP risk measure is a proposed solution to the limitations of the existing risk measures in capturing the specific risks associated with stop strategies, such as price volatility and the uncertainty of the exit time [14]. By accounting for these factors, the SPP risk measure aims to provide a more comprehensive and accurate evaluation of the risk for investors using stop strategies. This measure could be useful in helping investors make more informed decisions about their risk management strategies and potentially improve their overall portfolio performance.

In 1995, Konno and Shirakawa used an optimization approach to find the optimal stock portfolio with minimum semi-variance [15]. In the 1990s, the “value at risk” measure was proposed as another risk measure, which expresses the maximum potential losses an investor could face when choosing different assets or portfolios. The VaR model was first introduced as a means of managing risk in their own trading and investment portfolios [16]. Baumol (1963) introduced the idea of value at risk (VaR) for the first time, although it was not widely used until much later [17]. This was used while studying a model named “the confidence limit criterion of expected earnings,” which attempted to estimate the highest potential loss with a certain level of certainty. Artzner et al. (1999) introduced the concept of coherent risk measures, which are alternatives to the value at risk (VaR) measure [18]. They argued that VaR lacks two important properties: subadditivity and convexity. Mausser and Rosen (1998) showed that the VaR measure has multiple local minima in the optimization process [19].

The problem with local minima is that the optimization algorithm may not find the optimal combination of assets, leading to suboptimal portfolios and inaccurate risk estimates. VaR only considers the probability of a loss exceeding a certain threshold, but it does not account for the magnitude of the loss beyond that threshold or the tail risk associated with extreme events [18].

Rockafellar and Uryasev (2002) suggested a risk measure called the conditional value at risk (CVaR) as a way to address the limitations of VaR [20]. CVaR is a consistent risk measure that satisfies subadditivity and convexity, making it advantageous for dealing with securities. Its adoption can lead to more reliable risk management in securities markets [21].

Pflug and Swietanowski (2000), Rockafellar and Uryasev (2002), and Ogryczak and Ruszczynski (2002) have contributed to the development of the CVaR method from different perspectives [20,21,22]. Chekhlov, Uryasev, and Zabarankin, (2004) were the first to apply the CVaR minimization method to optimization problems [23]. An important consideration often overlooked in existing portfolio optimization methods is the investor’s exit time. To address this issue, the SPP-CVaR measure was developed by Bin (2015), which builds on the strengths of the CVaR measure and incorporates the exit time as a key factor [14].

This study investigated evaluations of the performance of assets when investors implement a stop strategy using the SPP-CVaR measure and Shannon entropy. In evaluating an asset’s efficacy, we utilized a model inspired by the range directional measure (RDM) model, which has been intricately adapted to cater to scenarios involving negative rates of return.

At the outset of our analysis, we computed the efficiency of an asset without considering its exit time, using the mean-CVaR model. Subsequently, we included the variable of exit time in our analysis and then added Shannon entropy to further enhance our approach. Calculation of the SPP-CVaR involves estimating the risk-neutral density using kernel density estimation, transforming it into the real-world density with a beta distribution, and calibrating the parameters. Price path simulations were generated, considering genuine market probabilities and risk. Real and SPP densities for prices and exit times were derived, incorporating predefined stop-profit points. Random entry times were simulated to improve the accuracy of the distribution of exit time. By applying the density transfer function, the SPP of the density of exit time was obtained.

Finally, the SPP-CVaR was calculated, focusing on critical investment exit times to quantify severe financial risk. Similarly, we calculated the mean return until the exit time, mirroring the process used for the SPP-CVaR. We then utilized the SPP-CVaR as an input and the mean return as an output in our model, enabling us to calculate the efficiency score of an asset. By incorporating the exit time and Shannon entropy, we found that the efficiency was more accurate. This enhancement could lead to several benefits, as we constructed a variety of portfolios catering to different time horizons, including short-term options (10 days, 20 days, 1 month, and 90 days), as well as a long-term portfolio.

These portfolios play a significant role in assisting investors and portfolio managers in their investment endeavors. By providing tailored investment strategies, they empower individuals to make informed decisions and effectively pursue their financial objectives. The incorporation of stop-profit points enables investors to secure profits and mitigate risks. Through the application of SPP-CVaR risk measures, we assessed the downside risk, facilitating informed allocation of the risk. Furthermore, incorporation of the exit time and the integration of Shannon entropy enhanced the decision-making efficiency. Efficiency scores were then utilized to optimize combinations of asset, aiding in the selection of suitable options. This comprehensive approach may empower investors to minimize risk, maximize returns, and make strategic choices aligned with their investment goals. Additionally, these measures provide valuable insights for company managers for informing risk reduction strategies and driving overall profitability.

The rest of this article is structured in the following manner. Section 2 presents the mathematical definitions and formulas. The methodology is explained in Section 3. Section 4 includes the experimental testing of the methodology and a comparison of the two models. Finally, Section 5 presents the conclusion.

## 2. Mathematical Definitions and Formulas

### 2.1. Risk Measures (VaR and CVaR)

Given a portfolio of n assets Φ={1,2,3,….,n}, the investment allocation of each asset is represented by the position vector $X=\left\{x_{1},x_{2},x_{3},\ldots x_{n}\right\}{\;}^{T}$, while the uncertain returns of the n risky instruments are expressed by the random vector $Y=\left\{y_{1},y_{2},y_{3,}\ldots ,y_{n}\right\}$.

$X^{T}Y$ can be used as a portfolio return to formulate $f\left(X,Y\right)=-X^{T}Y$, which is a more effective loss function.

Consider a fixed value x belonging to the set X, along with a probability distribution of y, The probability that the loss is less than or equal to a threshold α is defined as:(1)$$\psi \left(x,\alpha \right)=p\left(f\left(X,Y\right)\leq \alpha \right)=\int_{f\left(X,Y\right)\leq \alpha }\pi \left(x,\alpha \right)dy$$

The VaR associated with the portfolio Φ can be expressed, for a confidence level of β and any x∈X, as follows:(2)$$VaR_{\beta }\left(x,\pi \right)=min\left\{\alpha \in R:\;\psi \left(x,\alpha \right)\geq \beta \right\}$$

The conditional value at risk is equal to the average occurrence of losses greater than VaR at the confidence level of β.
(3)$$CVaR_{\beta }\left(x\right)=\frac{1}{1-\beta }E[-X^{T}Y\;|\;f\left(X,Y\right)\geq VaR_{\beta }]$$

The equivalent definition proposed by Rockafellar and Uryasev (2002) can be given as follows [20]
(4)$$CVaR_{\beta }\left(x\right)=\underset{\alpha \in R}{\min}\begin{array}{c} F_{\beta }\left(x,\alpha \right). \end{array}$$
where
(5)$$F_{\beta }\left(x,\alpha \right)=\alpha +\frac{1}{1-\beta }\int_{y\in R^{N}}\;{\left[f\left(X,Y\right)-\alpha \right]}^{+}\pi \left(y\right)dy$$
where π(y) is the notation for the probability distribution associated with Y.

### 2.2. SPP-CVaR Risk Measure

To calculate the SPP-CVaR, we begin with the price process, which is modeled by extending the model to include the exit time. The stochastic differential equation (SDE) used to describe the price process is based on geometric Brownian motion and is given by
(6)dS(t)=μS(t)dt+σS(t)dW(t)
where  μ and σ  are constants, and W(t) is the Wiener process or Brownian motion.

Let m  be the stop-profit point. The exit time for the stop-profit point m, denoted by $\gamma_{m}$, is given by the formula
(7)$$\gamma_{m\;}=min\left\{t\geq 0\;;W\left(t\right)=m\right\}$$
where $\gamma_{m\;}$ is the first time that the price reaches the stop-profit point. In practice, m is the difference between the buying and selling prices, subject to the investor’s considerations of the transaction costs and trading profits.

**Theorem** **1.***For*m∈ℝ+*, the cumulative distribution function of the time at which the price first reaches the stop-profit point was given by Bin (2015)* [14] *as follows*
(8)p{γm≤t }=22π∫|m|t∞e−y22dy  t≥0
*The density function is:*

(9)
$$f_{\gamma_{m}}\left(t\right)=\frac{d}{dt}\;p\left\{\gamma_{m}\leq t\;\right\}=\frac{\left|m\right|}{t\sqrt{2\pi t}}e^{-\frac{m^{2}}{2t}}$$


*The density function of the price process with a stop-profit point can also be obtained as follows:*

(10)
$$\pi \left(y\right)=\overset{n}{\underset{i=1}{\sum }}\frac{1}{\sqrt{2\pi \sigma^{2}\left(t-s\right)}}e^{-\frac{y_{i}{}^{2}}{2\sigma^{2}\left(t-s\right)}}$$


*The corresponding unconditional density function can be obtained from the conditional density function of the price process*

π(.)

*and the density function of the time that first crosses the stop-profit point*

$\;f_{\gamma_{m}}\left(.\right)\;$

*as follows:*

(11)
$$\nu \left(.\right)=\int_{0}^{T}\pi \left(.\right)f_{\gamma_{m}}\left(.\right)\mathrm{dt}=\overset{k}{\underset{i=1}{\sum }}\frac{1}{\sqrt{2\pi \sigma^{2}\left(t-s\right)}}e^{-\frac{y^{2}}{2\sigma^{2}\left(t-s\right)}}\frac{\left|m\right|}{t\sqrt{2\pi t}}\;e^{-\frac{m^{2}}{2t}}$$


*Finally, we can solve the arbitrage portfolio optimization problem by minimizing the SPP- CVaR*

(12)
$$\mathrm{Min}\;SPP-CVaR=\underset{\left(x,\propto \right)\in \chi *R}{\min}\alpha +\frac{1}{1-\beta }\int_{y\in R^{N}}{\left[f\left(x,y\right)-\alpha \right]}^{+}\;\nu \left(.\right)dy$$

*where*

(13)
$$\;\nu \left(.\right)=\int_{0}^{T}\pi \left(.\right)f_{\gamma_{m}}\left(.\right)\mathrm{dt}$$


*If the exit time *

$\gamma_{m}$

* follows a discrete distribution over time such as *

$\left\{t_{1}-k_{1},\ldots \ldots \ldots ,\;t_{n}-k_{n}\right\}$

*, since *

t>k

* and *

t=k+h

*, we have:*

(14)
$$\nu \left(y\right)=\begin{array}{cc} \overset{n}{\underset{i=1}{\sum }}\overset{n}{\underset{j=1}{\sum }}\frac{1}{\sqrt{2\pi \sigma^{2}h_{j}}} & e^{-\frac{y_{i}{}^{2}}{2\sigma^{2}h_{j}}} \end{array}\;\frac{\left|m\right|}{t\sqrt{2\pi t}}\;e^{-\frac{m^{2}}{2t_{j}}}$$



### 2.3. Shannon Entropy

Let η be a discrete random variable taking the values $a_{i}$ at the probabilities $p_{i}$, i=1, 2, 3,…., n. Its entropy was defined by Shannon and Weaver [24] as
(15)$$H\left(\eta \right)=-\overset{n}{\underset{i=1}{\sum }}p_{i}\;Ln\;p_{i}$$

Entropy is a measure of how evenly distributed the probabilities $p_{1},\;p_{2},\ldots p_{n}$ are. A higher value of entropy indicates that the random variable is closer to being equiprobable; in other words, each outcome is equally likely. Conversely, a lower entropy value suggests a deviation from this equiprobability.

The entropy value will reach its minimum of 0 if and only if there is an index k such that $p_{k}$ =1, and it will reach its maximum of ln n if and only if $p_{i}=1/n$ for all i=1, 2, ···, n.

The investment proportions in securities $x_{i}$ are treated similarly to the probabilities $p_{i}$ in the Shannon entropy formula, i.e., $x_{i}\geq 0$ and $\sum_{i=1}^{n}x_{i}=1,\;i=1,2,\ldots ,\;n$.

We substituted the probabilities of Shannon entropy with the proportions of investments and applied the principles of entropy to demonstrate the degree of diversification in a portfolio.

## 3. DEA-Based Evaluation Model of Assets’ Performance 

This section assesses assets’ performance in two distinct ways: first, over the entire time horizon using the CVaR risk measure, and second, up to the point of the investor’s exit time using the SPP-CVaR risk measure. The purpose was to compare performance between the two modes and assess the impact of the exit time on the assets’ performance. The subsequent subsections provide a detailed analysis and the results.

### 3.1. Evaluation of Assets’ Performance over the Entire Time Horizon Using Traditional Risk Measures

Based on the RDM model, the mean-CVaR model was proposed by Banihashemi and Navidi (2017) and can be described as follows [25]:(16)$$\begin{array}{c} \max\propto \\ S.t.\\ \;E\left(r\left(x\right)\right)\geq \mu_{o\;}+\propto R_{\mu_{o}}\\ \;CVaR\left(r\left(x\right)\right)\leq \;CVaR_{o}+\propto R_{CVaR_{o}}\\ \;\sum_{i=1}^{n}x_{i}=1\\ \propto \geq 0,\;\;0\leq x_{i}\leq 1,\;\;i=1,\;2,\;\ldots .,\;n \end{array}$$

In this model, μ is the expected return of the asset, $x_{i}$ is the proportion of the portfolio’s initial value invested in asset i, and x is an n-vector of variable $x_{i}$. Moreover, r is the return of the asset and $g=\left(R_{\mu_{o}},\;R_{CVaR_{o}}\right)$ is a vector that shows the direction in which ∝ is to be maximized. The process of determining the Mean−CVaR is similar to the RDM model. If the value of α assigned to the asset being evaluated is zero, then that asset is considered to be efficient. Alternatively, the efficiency of an asset can be represented as 1−α.

### 3.2. SPP-CVaR-Based Evaluation of Portfolios’ Performance

In this section, we introduce a model to evaluate the performance of an asset (or a portfolio) in a mean return–risk framework under the exit time and entropy. Unlike the previous model, this model considers the risks before the exit point, as exiting a transaction eliminates further exposure to the associated risks. Our model was inspired by the RDM model, with the risk (SPP-CVaR) and mean return considered as the only input and output, respectively, that use the uncertainty of the exit time. We used a sophisticated framework to determine the input and output of this model.

The computation of the stop-profit point–conditional value at risk (SPP-CVaR) is a comprehensive process that amalgamates statistical techniques, calibration methodologies, and simulation procedures. The overarching aim of this algorithm is to provide a robust framework for assessing the potential downside risks and tail outcomes associated with a given investment. This is achieved by taking both the real-world dynamics and the stochastic nature of the stop-profit points into account.

We began by obtaining the risk-neutral density through the application of kernel density estimation, a technique that captures the market’s sentiment and expectations by meticulously analyzing historical data. Subsequently, we utilized a beta distribution to transform this risk-neutral density into a real-world density. The parameters of a beta distribution that align with the risk-neutral density were determined using maximum likelihood estimation. We then computed the real-world density by applying the calibration function and the estimated parameters of the beta distribution.

Once the real-world density had been derived, we calibrated the real-world parameters through an optimization procedure. This procedure aimed to minimize a specific objective function and was designed to quantify the discrepancy between the real-world density and the density of the returns simulated from a normal distribution with predetermined drift and volatility parameters.

After calibration, we generated a price path simulation with the real-world measure. This measure was based on the genuine probabilities of events occurring in reality, and it also accounted for the inherent risk and uncertainty in the financial market.

In the subsequent step, we derived both the real and SPP density for the distribution of the price and exit time. This is a critical step for the computation of the SPP-CVaR. The real density reflects the true probability distribution of prices, while the SPP density integrates the concept of stop-profit points based on predefined profit levels. The stop-profit point was defined as a set that included 2%, 4%, 6%, 8%, 10%, and 12%.

Assuming that the investment entry time, represented as t, followed a uniform distribution, we executed a simulation of 250 random entry times. This simulation was crucial in deriving a more precise distribution of the exit times. The exit time for all 250 random entry times was computed to generate an exit time series, which aided in acquiring the distribution of real exit times using the kernel method. By implementing the density transfer function, we obtained the of density the SPP exit time.

The final stage of our process involved calculating the SPP-CVaR, a metric that quantifies the risk of severe financial loss linked to the stop-profit points. The SPP-CVaR is determined by the exit time, meaning it does not consider the full investment period. This approach offers a unique advantage, as it focuses on the critical moments of exiting from the investment, providing a more targeted and relevant measure of extreme risk, rather than diluting the risk assessment over the entire investment timeline.

In a similar vein to the SPP-CVaR, we also computed the mean return of the asset up until the point of the investor’s exit time.

Assuming that $Y_{day1},\;Y_{day2},\;Y_{day3},\;\ldots \ldots ,\;Y_{exit\;time\;}$ are the log of the returns of a specific asset up to the exit time and regarding the negative return, we defined the vector $g^{T}$ such that
(17)$$g^{T}=\left(R_{SPP-CVaR_{\beta }^{o}},\;R_{E\left(Y_{o}\right)}\right)$$
where
(18)$$\left(\begin{array}{c} \begin{array}{c} R_{SPP-CVaR_{\beta }^{o}}=\left[SPP-CVaR_{\beta }^{o}-\min\left(SPP-CVaR_{\beta }^{j}:j=1,2,\ldots ,n\right)\right]\; \end{array}\\ \;R_{E\left(Y_{o}\right)}=\left[\max\left(E\left(Y_{j}\right):j=1,2,3,\ldots ,n\right)-E\left(Y_{o}\right)\right] \end{array}\right)$$

This vector is a range of possible improvements in the input and output. The $SPP-CVaR_{\beta }^{o}$ is the value of the risk, and $E\left(Y_{o}\right)$ is the mean return of the asset. Then we solvef the following nonlinear model:(19)$$\begin{array}{c} \max\;\propto \\ S.t.\;E\left(Y\left(x\right)\right)\geq E\left(Y_{o}\right)+\propto R_{E\left(Y_{o}\right)}\\ \;SPP-CVaR\left(Y\left(x\right)\right)\leq \;SPP-CVaR_{\beta }^{o}-\propto R_{SPP-CVaR_{\beta }^{o}\;}\\ \;e^{T}x=1\\ \;-\sum xi\ln\left(x_{i}\right)\geq \;\theta \end{array}$$
where 0<θ≤ln(n), x≥0, ∝ ≥0.

The optimal solution $\propto^{*}$ indicates the inefficiency score of the asset under evaluation, and the asset is efficient when the inefficiency score is zero. The term for entropy was incorporated into the model conditionally, with θ ranging between zero and the natural logarithm of n. This proves beneficial when investors encounter numerous inefficiencies across a wide range of assets, making the construction of a portfolio challenging. Thus, entropy enhances efficiency, clarifying the investment process by aiding in the selection of the best assets from the available pool.

## 4. Empirical Application

In this section, we briefly introduce our data and present selective graphs of the distribution of the exit time, risk-neutral density, and real-world density. We then introduce short-term and long-term portfolios based on our analysis and research.

### 4.1. The Data Analysis

To highlight the capability of the proposed model in resolving the problem, it is crucial that we first introduce and discuss the data that have been collected for this purpose. The limitation of our study was the inability to include a larger number of assets in the portfolio due to the extensive empirical work required for each asset, potentially impacting the generalizability of our findings. Our portfolio comprised eight stocks, including Coca-Cola, Amazon, Pfizer, Tesla, crude oil, gold, Meta, and Bitcoin. We utilized historical daily closing prices from 11 February 2019 to 9 February 2023. All our data were sourced from the Yahoo Finance website. Due to the extensive numerical work involved, we were unable to include more assets in our portfolio. To ensure the accuracy of our calculations, we multiplied the return by 100. The decision to incorporate Bitcoin into our portfolio was influenced by Audit Rashid’s 2021 research [26], which indicates that the addition of Bitcoin could potentially boost a portfolio’s performance due to its potential for diversification, and its established resilience to market downturns offers an extra layer of protection during economic turbulence. In our analysis, we used Python to calculate the stop-profit point–conditional value at risk (SPP-CVaR). To evaluate the efficiency score of the asset, we used the general algebraic modeling system (GAMS). These tools allowed us to conduct our computations effectively and derive precise results.

As part of our methodology of computing the SPP-CVaR, acquisition of the distribution of the price and exit time proved to be crucial. Given the extensive data, we selectively included a limited number of graphs in this article for illustrative purposes. Figure 1, Figure 2, Figure 3 and Figure 4 showcase the risk-neutral density and the real-world density of selective assets. Additionally, Figure 5, Figure 6, Figure 7, Figure 8, Figure 9, Figure 10, Figure 11 and Figure 12 present the density of the SPP and the actual density of the distribution of the exit time for an asset, factoring in the stop-profit point.

Real-world density portrays the future asset price probabilities based on market data and real-world influences, while risk-neutral density simplifies valuation of the derivatives by assuming risk-neutrality among investors, aiding the efficient pricing of options and related instruments. As demonstrated by the figures, the probability density function of the exit times revealed a marked concentration of exit events during a particular trading day. As the exit time extended beyond this trading day, a noticeable and gradual decrease in exit probability became apparent, highlighting a distinct decline in the likelihood of these events.

### 4.2. Portfolio Selection

In this section, we evaluated the efficiency of all assets, considering scenarios both with and without the inclusion of the exit time and Shannon entropy. Subsequently, we construct short-term and long-term portfolios based on the exit time and Shannon entropy, SPP-CVaR, and efficiency scores, which are recommended for investors.

Table 1 provides an overview of the efficiency scored for the assets, not taking the time of exit and Shannon entropy into account.

As indicated in Table 1, Tesla and gold outperformed the other assets due to their higher efficiency scores.

In the subsequent section, we computed the exit time based on the stop-profit point. Following this, we determined the SPP-CVaR for the calculated exit time. Utilizing the SPP-CVaR and mean return, we assessed each asset’s effectiveness by evaluating its efficiency score in two ways: initially without factoring in the Shannon entropy and, subsequently, by incorporating various values of θ to encompass the influence of Shannon entropy. Ultimately, we constructed short-term and long-term portfolios based on our analysis. Our calculations were performed with a confidence level of 95%. Table 2, Table 3, Table 4, Table 5 and Table 6 show representations of both the short- and long-term portfolios we constructed.

An analysis of the efficiency score across three scenarios (without the exit time, with the exit time, and with both the exit time and Shannon entropy) can assist in selecting the optimal assets for portfolios. Instruments such as the exit time, SPP-CVaR, and the stop-profit point provide comprehensive perspectives. When low efficiency is a common issue among assets, Shannon entropy can boost the effectiveness of selection. For example, in Table 2, we present an assortment of assets suitable for inclusion in a short-term portfolio with a targeted period of 10 days. Investors can customize their selection based on their unique strategies, taking associated risks and predetermined exit timelines into account. With the exception of Pfizer, the efficiency scores of the other assets were relatively low, posing challenges for investors seeking the best asset options. However, by incorporating the entropy constraint, the assets’ efficiency can be enhanced, offering investors the opportunity to build more profitable portfolios.

In the domain of short-term investments, certain assets display strong performance, with the ability to generate rapid and consistent returns. For instance, in Table 3, Bitcoin’s exit time remained consistent for both the 2% and 6% stop-profit points, indicating its capacity to deliver prompt and reliable results. Moreover, in a 1-month portfolio, gold took merely 28 days to achieve a 2% stop-profit point and only 29 days for a 6% stop-profit point, emphasizing the importance of the exit time alongside the stop-profit point. These valuable insights underscore the significance of comprehending the exit times while designing short-term investment strategies.

For the decision on long-term investments, the relationship among when to exit, the efficiency scores, and risk is crucial. Looking at Table 6, we find interesting insights. At a 10% profit point, both Pfizer and Bitcoin had the same efficiency score of 1. However, Pfizer took 140 days to make that profit, while Bitcoin took 177 days. Despite the longer time, Bitcoin was less risky than Pfizer. At a 12% profit point, gold took more time to exit compared with oil and Pfizer, yet it had better efficiency and lower risk. Interestingly, Bitcoin’s efficiency score of 1 was even better than gold’s, and it also had less risk and quicker exit times. In conclusion, investors need to think carefully about these factors when making long-term investment choices.

If we compare the model that considers both the risk and return for the entire time horizon (as shown in Table 1) with our model, which takes the risk and returns only until the exit time into account, it is evident that our approach provides a more accurate representation. This is because, at the point of the investor’s exit, the consideration of risk and return becomes less relevant. Therefore, in our scenario, the efficiency score offers a more precise means of asset selection, resulting in a relatively dependable portfolio for the investor.

## 5. Conclusions

In conclusion, this study investigated the evaluation of the performance of assets using the SPP-CVaR measure and Shannon entropy, coupled with the data envelopment analysis (DEA) model for assessing the assets’ efficiency. Unlike other studies that consider the risk and mean return for the entire time horizon, our approach of incorporating the mean return and risk measures until the exit time significantly enhanced the accuracy of the efficiency scores. This enhancement facilitates the selection of the optimal assets for constructing a profitable portfolio. The analysis revealed that Shannon entropy enhances the effectiveness of asset selection, particularly in scenarios where low efficiency is prevalent among the assets. Developing effective investment strategies hinges on comprehending the dynamics among the exit time, risk measures, and efficiency scores. Whether one focuses on short-term or long-term investments, finding the right balance among these factors is essential for making the best asset choices. By taking this approach, investors and portfolio managers can minimize risk, maximize returns, and make strategic choices aligned with their unique investment goals, paving the way for more informed and successful financial endeavors. Our study’s results, originating from specific cases, have broader relevance across diverse situations. Further research can validate and expand upon these insights.

## Figures and Tables

**Figure 1 entropy-25-01252-f001:**
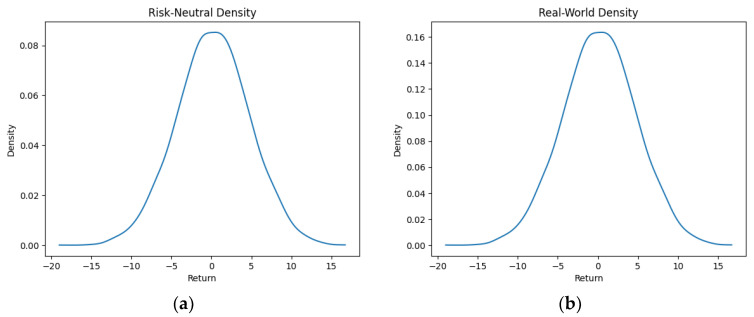
Risk-neutral density (**a**) and real-world density (**b**) of Bitcoin.

**Figure 2 entropy-25-01252-f002:**
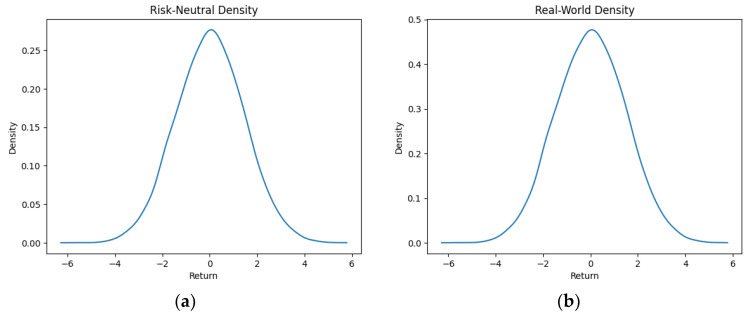
Risk-neutral density (**a**) and real-world density (**b**) of Coca-Cola.

**Figure 3 entropy-25-01252-f003:**
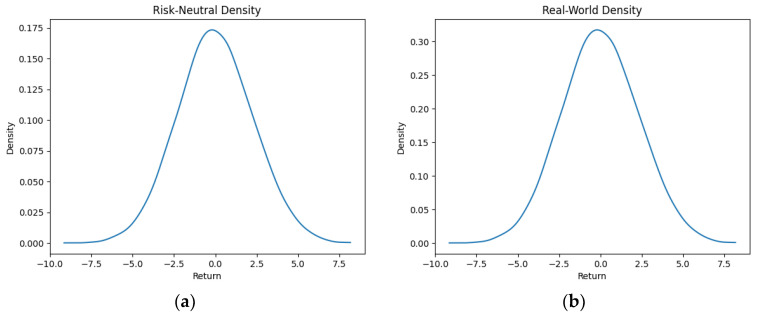
Risk-neutral density (**a**) and real-world density (**b**) of Amazon.

**Figure 4 entropy-25-01252-f004:**
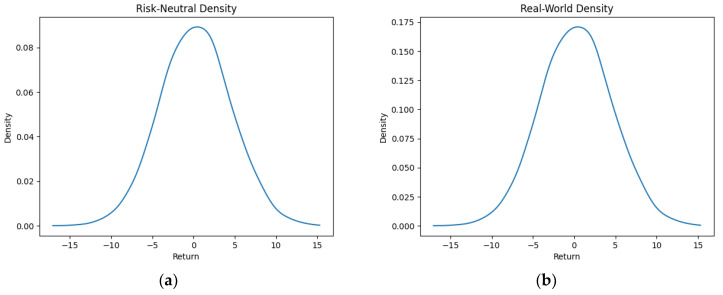
Risk-neutral density (**a**) and real-world density (**b**) of Tesla.

**Figure 5 entropy-25-01252-f005:**
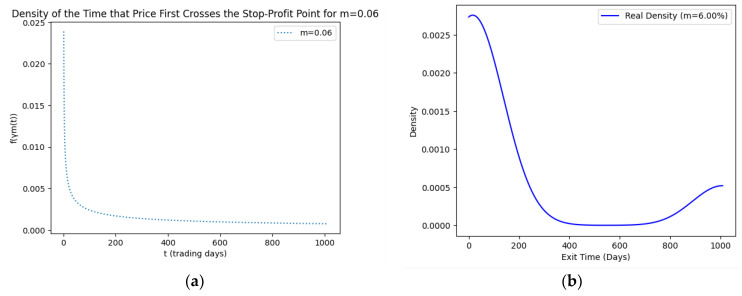
SPP density (**a**) and real density (**b**) of the distribution of the exit time of Coca-Cola (m = 6%).

**Figure 6 entropy-25-01252-f006:**
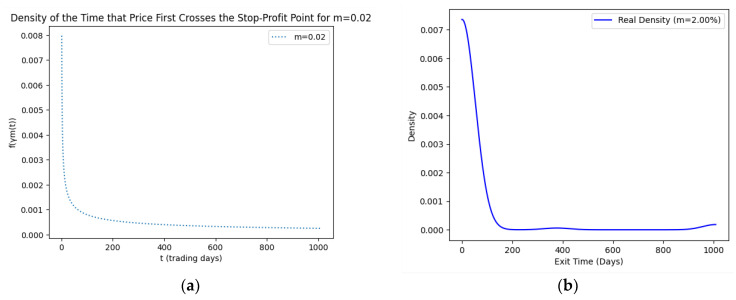
SPP density (**a**) and real density (**b**) of the distribution of the exit time of Amazon (m = 2%).

**Figure 7 entropy-25-01252-f007:**
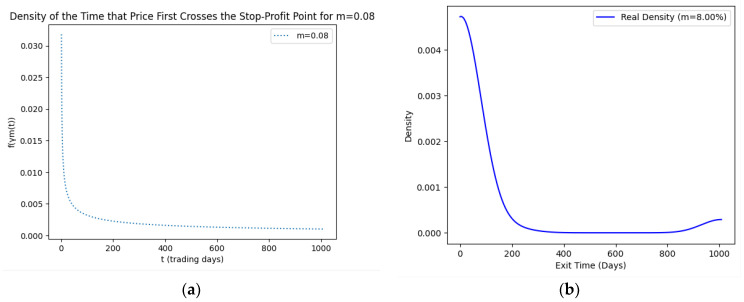
SPP density (**a**) and real density (**b**) of the distribution of the exit time of Tesla (m = 8%).

**Figure 8 entropy-25-01252-f008:**
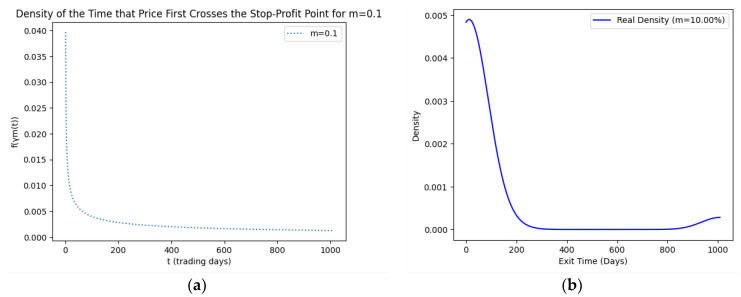
SPP density (**a**) and real density (**b**) of the distribution of the exit time of Pfizer (m = 10%).

**Figure 9 entropy-25-01252-f009:**
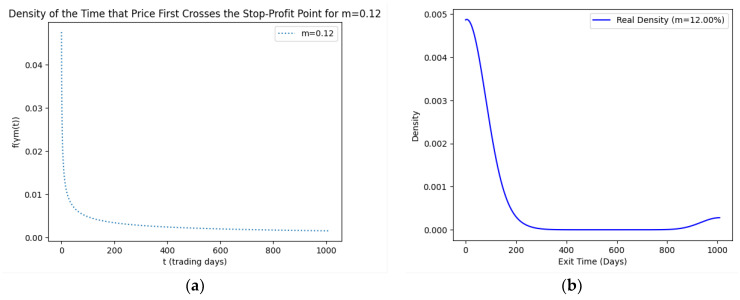
SPP density (**a**) and real density (**b**) of the distribution of the exit time of oil (m = 12%).

**Figure 10 entropy-25-01252-f010:**
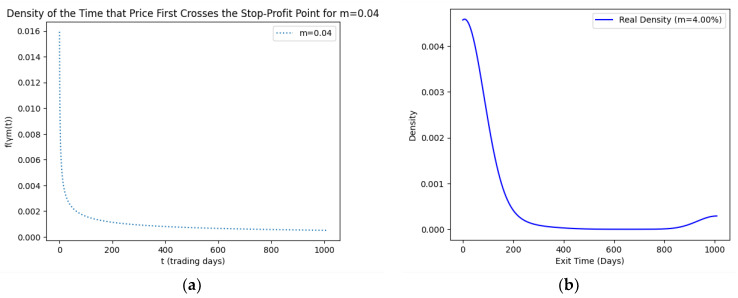
SPP density (**a**) and real density (**b**) of the distribution of the exit time of Gold (m = 4%).

**Figure 11 entropy-25-01252-f011:**
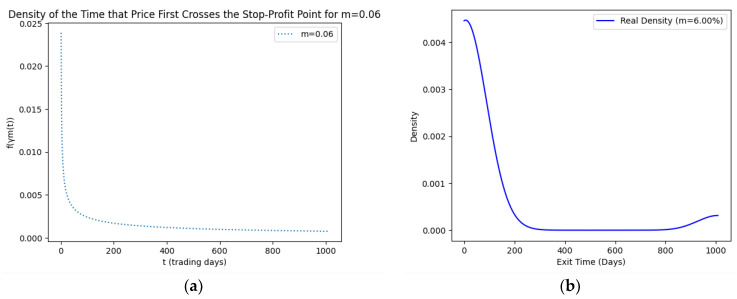
SPP density (**a**) and real density (**b**) of the distribution of the exit time of Meta (m = 6%).

**Figure 12 entropy-25-01252-f012:**
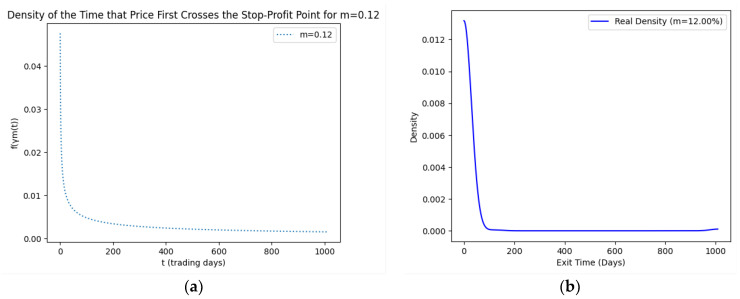
SPP density (**a**) and real density (**b**) of the distribution of the exit time of Bitcoin (m = 12%).

**Table 1 entropy-25-01252-t001:** Efficiency scores of assets independent of the exit time and entropy.

Asset	CVaR	Mean Return	Efficiency Score
Coca-Cola	0.03745	0.00018	0.79
Amazon	0.05365	0.00018	0.68
Pfizer	0.03969	0.0000865	0.75
Tesla	0.09969	0.00229	1
Oil	0.11004	0.00038	0.47
Gold	0.02453	0.00035	1
Meta	0.06882	0.00007469	0.58
Bitcoin	0.11175	0.00178	0.7

**Table 2 entropy-25-01252-t002:** Selection of assets suitable for a 10-day investment period.

Asset	Stop-Profit Point	Exit Time	SPP-CVaR	Mean Return	Efficiency Score with Exit Time	Efficiency Score with Exit Time and Entropy		
						θ = 0.75	θ = 1.5	θ = 2.2
Coca-Cola	2%	2	0.000230	0.00494	0.04	0.21	0.81	1
Coca-Cola	4%	8	0.0007	0.0031	0.02	0.14	0.36	0.76
Amazon	2%	8	0.000916	−0.0002	0.01	0.11	0.29	0.63
Amazon	6%	9	0.00096	0.0053	0.03	0.15	0.4	0.88
Pfizer	2%	1	0.00016	0.00461	1	1	1	1
Pfizer	4%	2	0.0001615	0.0189	1	1	1	1
Oil	2%	2	0.00064	−0.0229	0.01	0.06	0.18	0.63
Oil	4%	3	0.0012	0.01119	0.05	0.26	0.71	1
Meta	2%	8	0.00088	−0.0003	0.01	0.11	0.29	0.62
Meta	6%	7	0.000297	0.00453	0.03	0.19	0.54	1

**Table 3 entropy-25-01252-t003:** Ideal assets for short-term portfolio diversification over 21 days.

Asset	Stop-Profit Point	Exit Time	SPP-CVaR	Mean Return	Efficiency Score with Exit Time	Efficiency Score with Exit Time and Entropy		
						θ = 0.75	θ = 1.5	θ = 1.7
Coca-Cola	6%	16	0.001613	0.002759	0.05	0.09	0.3	0.47
Amazon	8%	10	0.0010479	0.00777	1	1	1	1
Amazon	10%	12	0.002868	0.00758	0.35	0.54	0.65	0.65
Meta	8%	14	0.001187	0.003991	0.27	0.28	0.5	0.68
Bitcoin	2%	21	0.0000577	0.00068	1	1	1	1
Bitcoin	6%	21	0.0000622	0.00267	1	1	1	1

**Table 4 entropy-25-01252-t004:** Comprehensive asset recommendations for a 1-month investment span.

Asset	Stop-Profit Point	Exit Time	SPP-CVaR	Mean Return	Efficiency Score with Exit Time	Efficiency Score with Exit Time and Entropy		
						θ = 0.75	θ = 1.2	θ = 1.5
Coca-Cola	8%	22	0.003711	0.002831	0.78	0.8	1	1
Coca-Cola	10%	24	0.003532	0.003301	1	1	1	1
Tesla	2%	26	0.001274	−0.0022	0.38	0.4	0.43	0.64
Tesla	6%	27	0.001165	0.0008237	0.82	0.84	0.9	1
Gold	2%	28	0.0008493	0.0005553	1	1	1	1
Gold	6%	29	0.000928	0.001130	1	1	1	1

**Table 5 entropy-25-01252-t005:** Beneficial asset choices for a 90-day investment timeframe.

Asset	Stop-Profit Point	Exit Time	SPP-CVaR	Mean Return	Efficiency Score with Exit Time	Efficiency Score with Exit Time and Entropy		
						θ = 0.75	θ = 1.2	θ = 2.01
Amazon	12%	96	0.01394	0.00114	0.19	0.21	0.48	0.94
Pfizer	6%	74	0.01018	0.000794	0.21	0.22	0.36	0.68
Pfizer	8%	79	0.01262	0.000767	0.18	0.19	0.34	0.66
Tesla	10%	74	0.00333	−0.000459	0.22	0.24	0.51	1
Tesla	12%	76	0.006018	0.00120	0.36	0.37	0.58	1
Gold	8%	43	0.001068	0.001377	1	1	1	1
Gold	10	98	0.0010646	0.000916	1	1	1	1
Meta	12%	43	0.004394	0.002012	1	1	1	1

**Table 6 entropy-25-01252-t006:** Long-term investment portfolio: suggested assets for sustained growth and stability.

Asset	Stop-Profit Point	Exit Time	SPP-CVaR	Mean Return	Efficiency Score with Exit Time	Efficiency Score with Exit Time and Entropy	
						θ = 0.75	θ = 1.5
Pfizer	10%	140	0.02169	0.000705	1	1	1
Pfizer	12%	142	0.02233	0.00068	0.82	0.85	0.92
Oil	8%	218	0.02490	0.000151	0.03	0.03	0.12
Oil	10%	219	0.01765	0.000428	0.05	0.05	0.2
Oil	12%	222	0.01308	0.000329	0.07	0.07	0.22
Gold	12%	363	0.000915	0.000324	0.94	0.94	1
Bitcoin	10%	177	0.000845	0.000205	1	1	1
Bitcoin	12%	179	0.000885	0.000542	1	1	1

## Data Availability

The data that support the finding of this study are available at https://finance.yahoo.com/, accessed on 9 February 2023.

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
