# Peer review of "The Effect of Exit Time and Entropy on Asset Performance Evaluation"

_entropy, 2023, doi:10.3390/e25091252_

Round 1

Reviewer 1 Report

The objective of this paper is to evaluate an asset performance using Shannon entropy. Authors use their methodology to create a portfolio aligned with short or long-term objectives.

I have some major concerns about this manuscript.

The introduction is excessively long and complex to read since the authors abuse excessively long paragraphs.

The literature review section as such, separate from the introduction section, is missing.

However, my major concern is the way in which authors develop the empirical application. First of all, there is not a methodology to choose the selected 8 stocks, therefore it could be thought that the authors have done it at their convenience.

Considering this sample used, the results are not significant. Conclusions are not clear and I do not see any relevant contribution to the financial literature in this paper.

What means these sentences:

“Our findings revealed that Bitcoin proved efficient for long-term investments, Pfizer for both short and long-term, and gold for short-term investments. Furthermore, our research also demonstrated the effectiveness of Shannon entropy, particularly in the context of a short-term portfolio.”

What is the utility of these results?

“In an environment characterized by significant inefficiency, it can be challenging for investors to select an asset.”

How authors now that the market is not efficient?

To conclude, it can be noted many paragraphs whose letter is different from the rest of the text, such as lines 99 to 103 or 281 to 285 and 289 to 290 among others.

Sometimes authors miss capital letters at the beginning of paragraphs.

The reading is dificult

Reviewer 2 Report

The study introduces a novel approach to asset performance evaluation using the SPP-CVaR (Stop Profit Point Conditional Value at Risk) and DEA (Data Envelopment Analysis) models. The study focuses on assessing asset performance over the entire time horizon as well as up to the point of exit time. It compares the results obtained from these models and evaluates the impact of exit time on asset performance. The methodology involves calculating risk measures, analyzing densities, and optimizing portfolios based on efficiency scores. The study utilizes historical data from eight assets and incorporates factors such as stop-profit points, exit time, and Shannon entropy for portfolio selection. The findings highlight the efficiency of certain assets, such as Tesla, Gold, and Pfizer, for different investment periods. Additionally, the study emphasizes the significance of Shannon entropy in enhancing portfolio selection in the short term. The study concludes by discussing the advantages of the proposed approach and suggesting potential avenues for future research.

 Referee Comment:

I have carefully reviewed the manuscript titled "The Effect of Exit Time and Entropy on Asset Performance Evaluation" and would like to provide my feedback to the authors.

The study offers several notable strengths:

1. Unique Approach: The study introduces a novel approach to asset performance evaluation by incorporating the concepts of exit time and entropy. By considering the exit time, the study acknowledges the practical aspect of exiting a transaction to eliminate further exposure to risks. The integration of Shannon entropy enhances portfolio selection by incorporating the degree of diversification and evenly distributed probabilities.

2. Comprehensive Evaluation: The study's use of the Stop Profit Point Conditional Value at Risk (SPP-CVaR) and mean return until the exit time provides a more comprehensive evaluation of asset performance. This approach goes beyond traditional models that assess risk over the entire investment horizon, allowing for a more realistic assessment of the risks associated with each asset.

3. Application of Data Envelopment Analysis (DEA): The use of DEA to calculate asset efficiency enables a comparative analysis among assets. This approach provides a nuanced understanding of asset performance and aids in identifying efficient assets for portfolio construction.

 While the study presents several strengths, there are also some limitations that should be acknowledged:

1. Limited Asset Selection: The study focuses on a relatively small number of assets, which may limit the generalizability of the findings to a broader range of assets. Including a more diverse set of assets would enhance the applicability of the study's findings.

2. Data Availability and Sample Size: The study relies on historical daily closing prices, sourced from a single platform, for the analysis. The availability and quality of data can significantly impact the accuracy of the results. Additionally, the limited sample size and specific time period may not capture all market conditions, potentially affecting the robustness of the findings.

3. Simplified Risk Measures: The study primarily uses the SPP-CVaR as a risk measure, which may not capture all dimensions of risk. Incorporating additional risk measures, such as tail risk or systemic risk, would enhance the comprehensiveness of the analysis.

4. Lack of External Validation: The study does not compare its methodology with other existing models or approaches, which limits the ability to assess the effectiveness and superiority of the proposed method. External validation would strengthen the study's findings and provide a benchmark for comparison.

5. Generalizability: The findings of the study may be specific to the selected assets, time period, and market conditions. It is important to acknowledge the limitations of generalizing the results to different contexts or timeframes.

 Despite these limitations, the study provides valuable insights into the effect of exit time and entropy on asset performance evaluation. I recommend that the authors address the limitations mentioned above and provide further discussion on the generalizability of their findings. Including a larger set of assets, validating the methodology externally, and considering additional risk measures would strengthen the study's conclusions. Additionally, discussing the potential implications and applications of the findings in real-world investment decision-making would enhance the practical relevance of the study.

Reviewer 3 Report

This paper studied a new approach to risk-return optimization considering the SPP-CVaR and mean return until the exit time.

The topic and results are interesting and valuable. 

I recommend the paper to be accepted for publication after minor revision.

1. 

Please, align the leading lines of equation (19).

2.

In Table 5 :  10 -> 10% 

Reviewer 4 Report

The abstract contains a lot about the methods, but it needs to focus more on what the novelty/innovation is in the paper and what the findings are.

The introduction provides a detailed discussion of various existing models, and how these existing concepts (such as SPP-CVaR) address various challenges, but it is very unclear as to exactly how the model proposed by the authors differs from existing models and what exactly the novelty or innovation is. The exact differences to prior studies need to be clearly summarised.

There needs to be more justification for choice of assets in the portfolio and for the time frame used.

The section on data should not be under a heading “empirical result” as data is an input, not a result.  

The biggest problem with this paper is the results section. It consists of a whole bunch of figures and tables with almost no discussion or interpretation of these. The reader is left with having to try and fathom out for themselves what it all means and what is new or different from what has been found before.  This section needs major work.

Only minor edits required

Round 2

Reviewer 1 Report

I cannot accept the publication of this manuscript because my main doubts have not been answered. The preselection method is excessively biased and therefore there is no way to ensure that the authors have not chosen those stocks that perform best according to the presented methodology.

Reviewer 2 Report

 Accept after corrections to minor errors and text editing.

 Accept after corrections to minor errors and text editing.

Reviewer 4 Report

The revisions are satisfactory.

Mostly fine. Just needs minor checking.

Round 3

Reviewer 1 Report

All my concerns have been answered correctly.